# Implementation of a Single Emulsion Mask for Three-Dimensional (3D) Microstructure Fabrication of Micromixers Using the Grayscale Photolithography Technique

**DOI:** 10.3390/mi11060548

**Published:** 2020-05-29

**Authors:** Intan Sue Liana Abdul Hamid, Beh Khi Khim, Sofiyah Sal Hamid, Mohamad Faizal Abd Rahman, Asrulnizam Abd Manaf

**Affiliations:** 1Collaborative Microelectronic Design Excellence Center (CEDEC), Universiti Sains Malaysia (USM), Sains@USM, Pulau Pinang 11900, Malaysia; liana@uthm.edu.my (I.S.L.A.H.); behkhikhim@student.usm.my (B.K.K.); sofiyah@usm.my (S.S.H.); faizal635@uitm.edu.my (M.F.A.R.); 2Faculty of Electrical and Electronic Engineering, Universiti Tun Hussein Onn Malaysia, Parit Raja, Johor 86400, Malaysia; 3Faculty of Electrical Engineering, Universiti Teknologi MARA Cawangan Pulau Pinang, Pulau Pinang 14000, Malaysia

**Keywords:** grayscale photolithography technique, micromixer, mixing performance, 3D microstructures

## Abstract

Three-dimensional (3D) microstructures have been exploited in various applications of microfluidic devices. Multilevel structures in micromixers are among the essential structures in microfluidic devices that exploit 3D microstructures for different tasks. The efficiency of the micromixing process is thus crucial, as it affects the overall performance of a microfluidic device. Microstructures are currently fabricated by less effective techniques due to a slow point-to-point and layer-by-layer pattern exposure by using sophisticated and expensive equipment. In this work, a grayscale photolithography technique is proposed with the capability of simultaneous control on lateral and vertical dimensions of microstructures in a single mask implementation. Negative photoresist SU8 is used for mould realisation with structural height ranging from 163.8 to 1108.7 µm at grayscale concentration between 60% to 98%, depending on the UV exposure time. This technique is exploited in passive micromixers fabrication with multilevel structures to study the mixing performance. Based on optical absorbance analysis, it is observed that 3D serpentine structure gives the best mixing performance among other types of micromixers.

## 1. Introduction

The advances in microfluidic technology is continuously expanding since the first invention of micro-total analysis system (µTAS) [1]. During this phase, early microfluidic devices were fabricated using IC-based microfabrication technology. The IC-based fabrication technique known as photolithography only enables the design of two-dimensional (2D) planar structures. Meanwhile, the three-dimensional (3D) structures with various depth profiles channel features are emerging in microfluidic development including the biomedical, biochemical and bioanalysis devices [2,3,4]. Though, several techniques have been reported for 3D microstructures fabrication. The layer-by-layer technique was introduced using the same 2D photolithography fabrication setup. Applying the concept of multiple coatings and exposures, the potential alignment errors increase, and this technique becomes difficult to be implemented [5]. Electron beam lithography (EBL) offers 3D microstructures fabrication with better resolution than the conventional photolithography. The 3D microstructures are created by electron beam process across the resist’s surface. The resultant depth of the resist can be varied by applying variable dose of electron beam [6,7]. The same 3D microstructure created by EBL can also be produced by using laser direct technique using a focused laser beam, but with lower resolution [8]. These techniques require an expert to operate with high cost equipment maintenance. In addition, the slow point-to-point and layer-by-layer writing process will result in low throughput and not suitable for a large surface area fabrication. These issues thus give limitations for their application in microfluidic devices fabrication [9,10]. An unconventional 3D microfabrication method such as multidirectional lithographic is also possible using techniques such as incline, rotate and back exposure by UV light. This method, however, requires special fabrication setups for precise angle control either by moving the UV source or the substrate stage [11]. X-ray lithography is another method for 3D microstructure fabrication with precision but suffers from restricted radiation source [12]. In line with industrial revolution 4.0, 3D printing becomes another attractive approach for 3D microfabrication. The technology enables the construction and fabrication of 3D structures without tedious and multiple fabrication steps as in the conventional process. The technique is, however, still limited by the issues of speed, precision and micro size resolution [13,14]. In recent years, the use of gray scale fabrication technique starts to gain attention in LOC fabrication due to its advantages of low-cost and one-step fabrication technique. The use of this technique has been successfully implemented to fabricate several functional structures in microfluidic’s application such as microlens, cantilevers and microneedles [15,16,17]. The technique has capabilities to produce arbitrary shape with high lateral and vertical resolution [18]. The use of one-step maskless photolithography technique also demonstrates several advantages in terms of its simplified process, time and cost saving as well as high throughput [19].

The implementation of these technique in a variety of 3D structures of micromixers is yet to be explored, which is the motivation of this work. Micromixer is one of the important structures in microfluidic system to perform the mixing process of several liquid before being transported to the next stage for sensing and sorting application [20,21,22,23]. In previous works, different structures of the micromixers’ channel have been proposed based on the traditional fabrication process [24,25]. The exploitation of 3D multilevel structures in micromixers gives significant impact on mixing performance which could affect the overall performance of the proposed device [26,27,28]. 

In this work, the capabilities of the grayscale photolithography technique in fabricating passive micromixers with multilevel microstructures are demonstrated. Through this process, four types of 3D multilevel structures are proposed. The performance of the mixing process for each type will be measured using suitable techniques such as photometric and spectroscopy to identify the type of micromixer that offers the best mixing performance. 

## 2. Methodology

In general, the work is divided into three stages. Firstly, the calibration of the grayscale photolithography technique based on grayscale thickness profiling is performed by using a negative photoresist (SU8) on a glass substrate. Secondly, the grayscale photolithography is implemented to develop different types of 3D multi-structure of passive micromixers, which aims to overcome the limitation in the current fabrication approach. Finally, the characterisation of mixing profile of passive micromixers is performed using the standard characterisation methods.

### 2.1. Grayscale Thickness Profiling

The calibration of variable grayscale concentrations for various height profiles is needed prior to the 3D microfabrication of passive micromixer. The process begins with the emulsion mask (High Precision Photo Plate from Konica Minolta Inc., Tokyo, Japan) preparation. Emulsion mask was prepared by using optical projection lithography using software mask as the master mask. The software mask was printed on a transparent-based polyethene terephthalate (PET) film. Then, the image on the software mask was transferred onto emulsion mask by using the Simple Mask Fabrication Machine (MM605 from Nanometric Technology Inc., Tokyo, Japan) in a 5-to-1 scaling down ratio. The software design mask and emulsion mask were shown in Figure 1a,b, respectively. In this paper, SU8-10 was selected because initial target micro 3D structure thickness is between range 15–50 um height before modification on lithography grayscale exposure. In order to remain the thickness of SU8, the spin coating process of SU8 was skipped in this method. Using a low-viscosity SU8 is better for the SU8 to flow on the overall surface of the glass substrate during the baking process with control by surface tensioning. The SU8 sample was prepared by dispensing 2.5 mL volume of SU8-10 on a clean 3 × 1 inch microscope glass slide from DURAN group. The coated glass substrate underwent a soft baking process at 95 °C for 7 h on a hotplate. Then, the coated glass substrate was aligned with the emulsion mask for UV light exposure. The exposure process was carried out using One Side Mask Aligner (LA4100_R1 San-Ei Giken Inc., Hyogo, Japan) with a power density 180 W of the i-line mercury lamp. Then, the exposure variation of exposure time will be characterized to achieve optimised maximum thickness of SU8-10 during grayscale lithography technique. This work is to prove the capability of SU8-10 in fabrication different height of 3D microstructure SU8 mould by a single emulsion mask. The photoresist SU8-10 was exposed for 30 and 60 s by back exposure method. Then, the samples were baked at 65 °C for 2 min, then at 95 °C for 10 min. The samples were cooled down gradually into room temperature before development process using SU8 developer. The developed samples have multilevel thickness as shown in Figure 1c. The thickness was measured using a manual Coordinate Measuring Machine (CMM) from Mitutoyo (Kanagawa, Japan).

Then, the data of SU8 thickness against grayscale concentration is obtained and plotted to indicate the relationship between these variables for different periods of UV exposure. The data is used to select the right depth or thickness of the SU8 mould during designing stage.

### 2.2. Application of the Grayscale Fabrication Technique for Passive Micromixer Development 

Passive micromixers were developed using the proposed grayscale fabrication technique. In this work, the conventional photolithography setup is used for 3D microfabrication by grayscale photolithography technique. However, the use of binary mask is eliminated due to the all-or-none exposure scheme. Instead, binary mask is replaced with grayscale photomask that offers a gradient dose illumination of exposed areas using a single exposure. This will lead to the generation of microstructures with various thickness profiles (single mask). The concept of grayscale photolithography is described in Figure 2. Grayscale photomasks such as High Energy Beam Sensitive (HEBS) glass or thin film metallic mask that are normally used in grayscale fabrication are costly because the masks are generated using scanning lasers and electron beam. In this paper, a simple 3D microfabrication method by combining an emulsion grayscale photomask and standard soft lithography technology is described. The emulsion grayscale photomask is generated by optical projection lithography to maintain cost effectiveness and less complexity of the fabrication process. Backside exposure has been used to create curving structures with different height profiles with bell-shaped cross section. 

Four types of micromixer with different structures were designed and fabricated. These are the planar channel, multistep channel, B-spline curve channel and 3D serpentine micromixer. In general, the length and width of mixing channels for all micromixers were designed at 40 and 600 µm, respectively. The depth of the mixing channel and the inlet/outlet wells is 500 and 1 mm, respectively. The planar channel micromixer serves as a reference for mixing performance evaluation, where the flow was assumed to be in smooth flow without any obstacles. As for others, 16 mixing units with obstacles were designed and introduced inside the microchannel of each type. For the multistep channel, each mixing unit consists of multi steps obstacles with four different height and for B spline, obstacles were designed based on criteria discussed by [29]. Meanwhile, for 3D serpentine, the obstacles were formed by combining two layers of microchannels based on the concept introduced by [30,31].

Figure 3 shows the software mask design for all micromixers. Then, the SU8 mould was developed by using the technique that has been described earlier. The SU8 mould, as shown in Figure 3, was replicated by (Polydimethylsiloxane) PDMS to create a PDMS-based micromixer. This technique is also known as soft lithography. The PDMS mixture was prepared with 10-to-1 ratio of the polymer base and curing agent (Sylgard 184, Dow Corning, Midland, MI, USA). Then, the mixture was poured onto the SU8 mould and degassed in vacuum chamber for 20 mins. After the degassing process, the sample was placed in an oven at 120 °C for 1 h. After 1 hour, the PDMS was solidified and was lifted-up to obtain the PDMS microchannel. Another layer of flat PDMS was punched with inlet and outlet wells and then a thin layer of PDMS mixture was spin coated on it. Finally, both layers were aligned and sealed together to create a close microchannel of the micromixer as shown in Figure 3. The PDMS-based micromixer was degassed and cured in an oven at 120 °C for 1 h. The sealing process using PDMS solvent. The bottom PDMS will be spin coated with wet PDMS, the pattern area will be covered using a tape. Then, the tape will be peel off after the coating, then the top PDMS was will adjusted to bottom PDMS to ensure the position of the pattern. Then, the completed seal PDMS–PDMS was baked in an oven to strengthen the sealing process [32]. 

### 2.3. Mixing Performance Evaluation 

Finally, the mixing performances of the micromixers were characterized by mean of photometric and spectroscopy analysis experiment. For both techniques, two solutions were used to study the mixing performance: red dye solution and distilled water. Red dye solution was prepared at 1.00 mg/mL concentration and considered as having the highest concentration, while distilled water (0 mg/mL of red dye) is considered as to be the lowest. Red dye solution and distilled water were injected simultaneously into the inlets with a 1.5-mL/min flow rate. Both solutions were injected into the micromixer inlets through tygon tubes using a digital syringe pump (New Era Pump Systems Inc., New York, NY, USA). For, photometric analysis, the mixing performance was interpreted visually by using inverted microscope (Nikon eclipse TS100, Manufacturer, Tokyo, Japan) and displayed at the computer monitor as depicted in Figure 4.

In order to quantify the mixing performance, the spectroscopy technique was used. Figure 5 illustrates the schematic diagram of the off-chip absorbance measurement. The droplet from the outlet of micromixer was transferred into a micro-drop plate reader (SkanIt Go by Thermo Scientific, MA, USA) and measured by spectrometer. The pathlength for the absorbance measurement was set at 5 mm. 

The performance is evaluated based on how well the red and distilled water were mixed together through the red colour degradation/decolourisation. This technique relies on the correlation between the absorbance and concentration of red dye colour, which is also representing the colour degradation of the mixed solution. As mixing occurs, the colour of red dye solution is getting lighter as well as the concentration of the dye. According to Beers law, as the concentration of the solution (also reflected by its colour) decreases, the absorbance also decreases, and vice versa. Thus, the concentration measured at the outlet by absorbance is considered to represent the mixing behaviour through its colour degradation. This means the absorbance from spectroscopy measurement is suitable to quantify the mixing behaviour (through its changing colour/colour degradation), which is directly correlated to red dye’s concentration. 

In this work, two solutions were used as calibration points: a red dye solution prepared at 1 mg/L (absorbance of A) was considered as the upper limit value and distilled water with 0 mg/L of red dye (absorbance B) was considered as the lower limit value. These two values were used to calculate the threshold value, C, which representing the complete mixing. Figure 6 illustrates the concept used for quantifying the mixing process.

It is clear that the mixing of these two solutions (upper limit and lower limit) will produce a degradation in concentration and its red colour (red colour becomes lighter), so as the absorbance. For the best (complete) mixing, it was assumed that the mixture at the outlet would give 50% of red dye and 50% of distilled water, which produced the concentration of 0.5 mg/L of red dye. If the Beers law is obeyed, the absorbance value of the complete mixing would be halved and calculated as C, a threshold that represents the concentration of 0.5 mg/mL (which can also be calculated theoretically). Using these two values of upper and lower limit of absorbance values. In determining the best micromixer design, the mixing performance was evaluated based on the closest value towards the threshold value, C. The structure with capability of producing the best mixing (i.e., absorbance equals or closest to C) at the outlet was considered to be the best structure for mixing.

## 3. Results and Discussion

### 3.1. Grayscale Concentration–Thickness Relationship

Figure 7 shows the plot of developed SU8 thickness against grayscale concentration for both UV exposure time. In general, it is observed that for both UV exposure times, the concentration of grey level had successfully developed a relative SU8 thickness as result of the photolithography process. As clearly seen in this graph, the SU8 thicknesses obtained from 60 s UV exposures are higher than the 30 s UV exposures. It indicates that, at specific grayscale level, the longer the photoresist has been exposed to UV light, the higher the thickness of SU8 can be developed, which is in agreement with the initial expectation.

From the plot, the maximum average of SU8 thickness achieved is 1108.7 µm with 60 s UV exposure time while the minimum was recorded as 242.8 µm for the same exposure time. When the exposure time is reduced to 30 s, the maximum average SU8 thickness was recorded as 774.8 µm while the minimum was 163.8 µm. These results revealed that higher UV exposure dose (grayscale level and exposure time) created photoresist with higher structure. The graph also show that different UV exposure times give different effects on the initial structure to be developed. This is observed as no result was achieved for grayscale concentration below 60% and 52% when exposed with UV for 30 and 60 s, respectively. For a 30-s UV exposure time, the SU8 thickness starts to be developed when the grayscale level is at 60%. Meanwhile, for 60 s UV exposure time, the SU8 thickness starts at 52%. This indicates that, at lower concentrations, the exposure time needs to be increased to realise a lower structure.

In terms of linear correlation between grayscale concentration and SU8 structural height, the plots are analysed for the concentration range between 60%–98%, where both plots show a considerable structural height. Figure 8 shows the graph with linearity lines and error bars (5%).

The coefficient of correlations are computed to be 0.9958 and 0.9837 for both exposure times of 30 and 60 s, respectively. The high linear relationship of the plots shows the reliability of this technique in developing SU8 at a specific height. Meanwhile, the vertical resolution of the technique is represented by the gradient of the plot, which are found to be quite similar at 33 μm for each grayscale level (%). These results are used as a guideline in the fabrication of passive micromixers by taking into account the targeted thickness, grayscale concentration and the UV exposure time.

### 3.2. Fabrication of Passive Micromixer

This section discussed the result of fabrication of micromixer based on the SEM images obtained for SU8 structure (vertical dimension) and PDMS chip (lateral dimension). Figure 9 shows the example of SEM images of the SU8 mould fabricated using the grayscale fabrication technique. The targeted height for both structures is 500 μm, which corresponds to the grayscale concentration of 64% (at 60 s UV exposure) as obtained from the grayscale calibration table. The images are taken at a 45° tilt angle for the SU8 mould of two micromixers’ structure, planar and B spline.

The images indicate that the obtained heights of the mixing channel for planar and B spline structures are 518 and 506 μm, respectively. These heights are approximate according to the targeted design with tolerances of less than 5%, thus verifying the effectiveness of the grayscale technique in the fabrication of SU8 mould. Figure 10 shows the SEM images of the PDMS micromixers with various geometry after being peeled off from the SU8 mould through the soft lithography process. The images are used to study the lateral dimension of the PDMS chip resulting from the process.

As the targeted microchannel width is 600 μm for all types, it can be observed that the structure can be realized with some acceptable tolerances. Table 1 shows the summary of deviation error for these types of micromixers as observed from Figure 10, where the actual width of the microchannels are slightly varies with deviation error of less than 7%.

Overall, the SEM images used to characterise the fabricated structure of micromixers’ mould and chip indicate the capability of the technique to be implemented in producing vertical and lateral dimensions of 3D structure of micromixer, with very low deviation errors. These errors are mostly due to the complexity of the structure itself and the degradation quality of image transfer during photolithography process such as line shortening and corner rounding effect as observed for 3D serpentine type.

### 3.3. Mixing Performance

Figure 11 shows the photometric result taken at three different points of various structure of micromixers under test. At this stage, the mixing performance is evaluated based on visual observation at the point of interest (*z* = 40 mm). 

Figure 11a shows that, for the planar channel micromixer, there is an obvious separation between both solutions (at the end of mixing channel), indicating the ineffective mixing of the structure since the mixing only relies on diffusion. For the multistep micromixer, it shows an improved mixing as shown in Figure 11b. The multilevel structures inside the microchannel promotes continuous stretching and folding of the fluid stream which results in better mixing. However, apparently at the end of the mixing channel both fluids still do not mix completely. In Figure 11c, the B-spline micromixer shows a better mixing visualisation as compared to the previously discussed micromixers. In this micromixer, the resulting good mixing is caused by continuous stretching and folding and the split and recombined of the fluid stream at the obstacles inside the B-spline curve. Finally, the best mixing is observed for the 3D serpentine micromixer as shown in Figure 11d. At the end of the mixing channel, obviously the red colour dominates which indicates good mixing. The two crossing layers channel of the micromixer has generated chaotic advection by stretching and folding, splitting and recombining, and the increasing contact area between channel layers offers a better diffusion of fluids’ flow. 

Spectrometry analysis was carried out to evaluate the mixing performance of different structures of micromixers under test based on the relationship between absorbance and concentration of red dye. Figure 12 shows the absorbance spectra of two calibration points, i.e., 1 mg/mL red dye and distilled water (0 mg/mL concentration of red dye), which are to be used in determining the mixing performance at the outlet of these structures. In order to reduce the measurement error, there are five measurements taken for each point.

In general, it is observed that the peak wavelength occurs at 515 nm, indicating that for the red-colour solution, this wavelength is the best option if the monochromatic light source is used. As shown in the figure, five different values of absorbance were observed for the highest concentration of red dye solution (1mg/mL of red dye) with the average of 1.38. Meanwhile, at the lowest concentration (0 mg/mL of red dye), the absorbance for the five measurement samples are consistent at 0.08.

In theory, if the mixture obeys Beers law, complete mixing between red dye solution and distilled water will produce the absorbance value of 0.65, which corresponds to the concentration of 0.5 mg/mL of red dye. This value is used as a threshold to determine the mixing performance of different structures of micromixers. The closer the absorbance towards 0.65, the better the mixing between two solutions. Table 2 summarizes the finding that is used as calibration points for evaluation of mixing performance.

At this peak wavelength, the absorbance is used to quantify the mixing performance of a mixture between red dye solution (highest concentration) and distilled water (lowest concentration). From the absorbance measurement at the outlet of each structure, it is found that the absorbance values at these outlets are 1.09, 0.85, 0.78 and 0.74 for planar-channel, multistep, B-spline and 3D serpentine micromixers, respectively. According to the concentration calculation, these absorbances correspond to the mixing percentages of 81.3%, 63.4%, 58.2% and 55.2%, respectively. From these values, it can be concluded that the best mixing is achieved for 3D serpentine micromixer with an absorbance value closest to 0.65 (representing complete mixing as in Table 2), which is also in agreement with the results obtained from the photometric approach. 

## 4. Conclusions

The grayscale fabrication technique with backside exposure using negative photoresist SU8 has been successfully demonstrated in this work by using a single mask with single exposure. The approach has the advantages of low cost and rapid fabrication process. At two different exposure times, 30 and 60 s, the 3D microstructures with various heights between 163.8 and 1108.7 µm are possible with a grayscale concentration between 62% to 98%. The implementation of this technique in fabricating different structures of passive micromixer verifies its effectiveness. Different structures of passive micromixers show different mixing performance due to the introduction of obstacles fabricated inside the microchannel. Such obstacles oppose the liquid flow and causes different flow behaviours which consequently affect the mixing process of the two different solutions.

## Figures and Tables

**Figure 1 micromachines-11-00548-f001:**
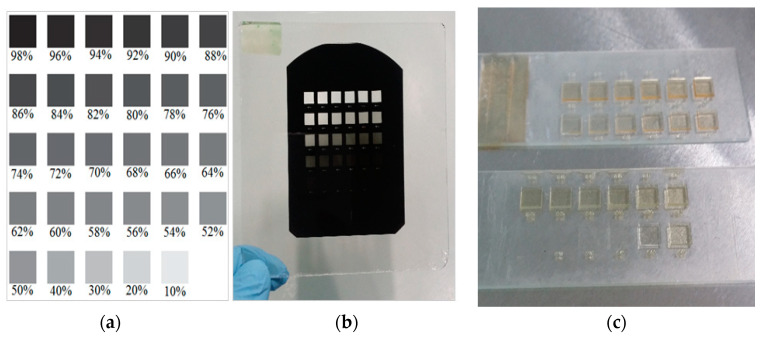
(**a**) software mask, (**b**) emulsion mask and (**c**) SU8 mould of various grayscale concentrations.

**Figure 2 micromachines-11-00548-f002:**
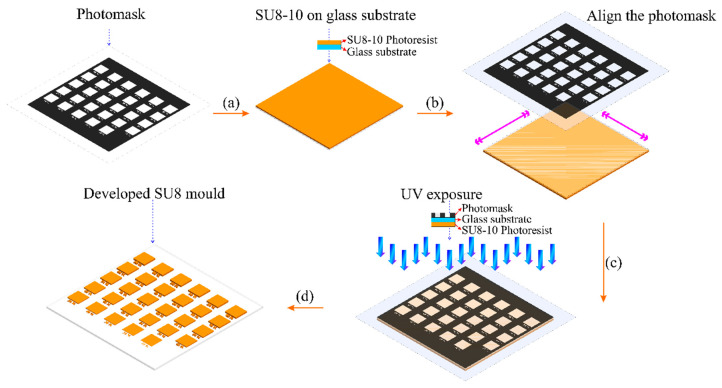
Grayscale photolithography process for SU8 mould development.

**Figure 3 micromachines-11-00548-f003:**
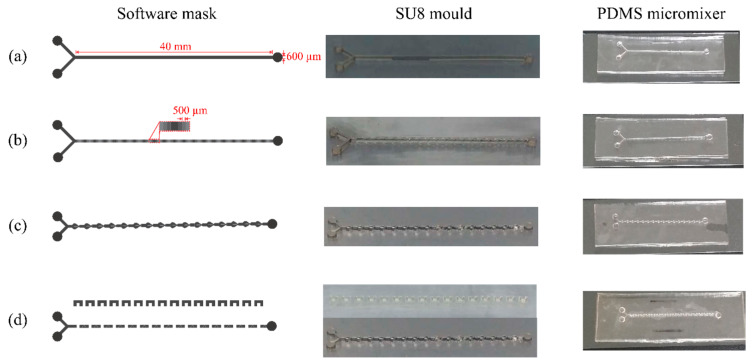
The software mask, SU8 mould and PDMS micromixer for: (**a**) planar, (**b**) multistep, (**c**) B-spline and (**d**) 3D serpentine micromixers.

**Figure 4 micromachines-11-00548-f004:**
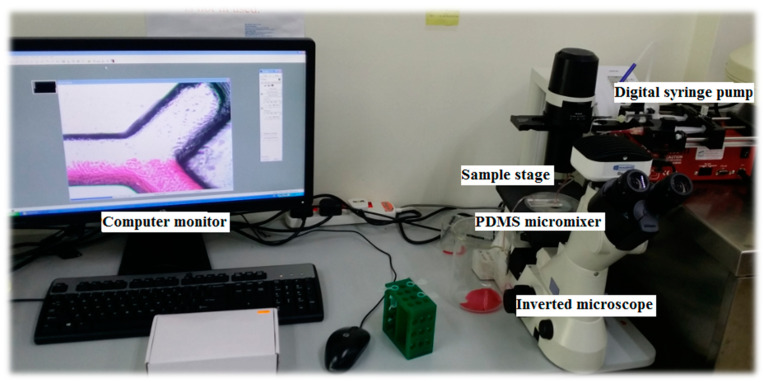
Experimental setup for photometric experiment.

**Figure 5 micromachines-11-00548-f005:**
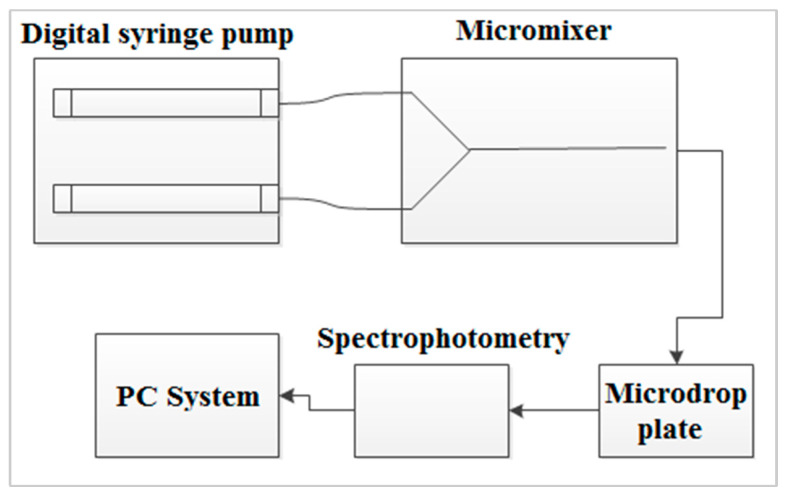
Off chip absorbance measurement.

**Figure 6 micromachines-11-00548-f006:**
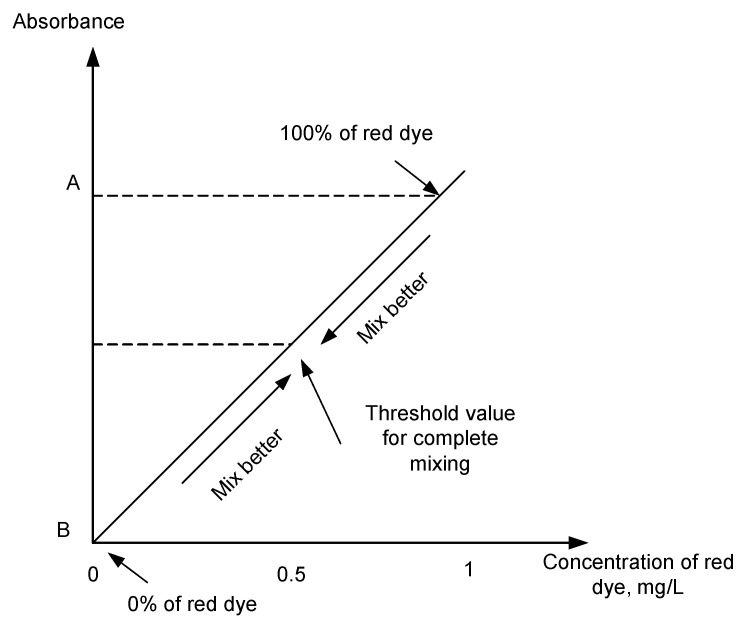
Concept of quantification of mixing performance.

**Figure 7 micromachines-11-00548-f007:**
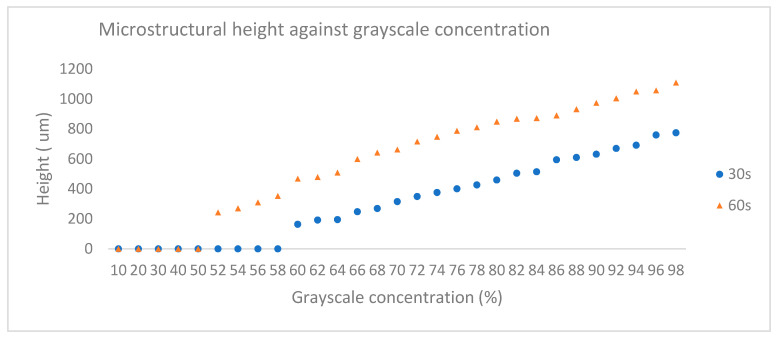
Photoresist thickness versus grayscale concentration (10%–98%) plot for 30 and 60 s UV exposure time.

**Figure 8 micromachines-11-00548-f008:**
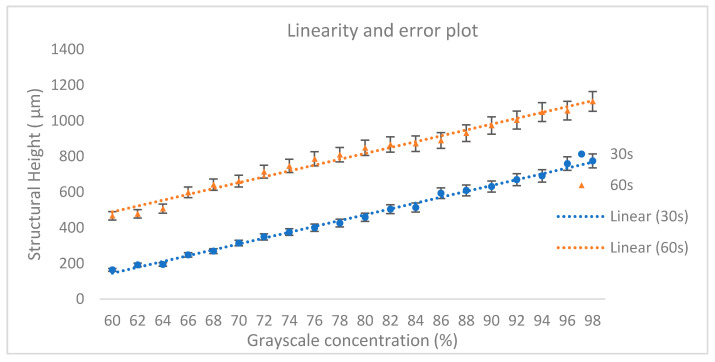
Linear plot at grayscale concentration between 60%–98% for 30 and 60 s UV exposure time.

**Figure 9 micromachines-11-00548-f009:**
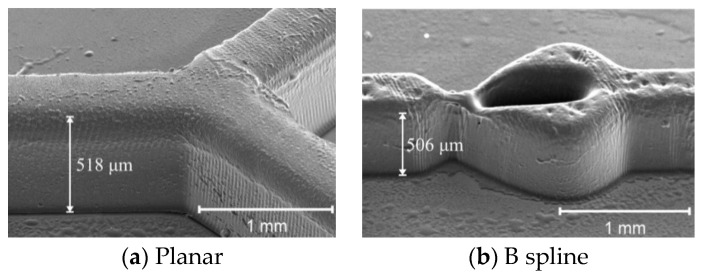
SEM images of 45° tilt angle of SU8 mould.

**Figure 10 micromachines-11-00548-f010:**
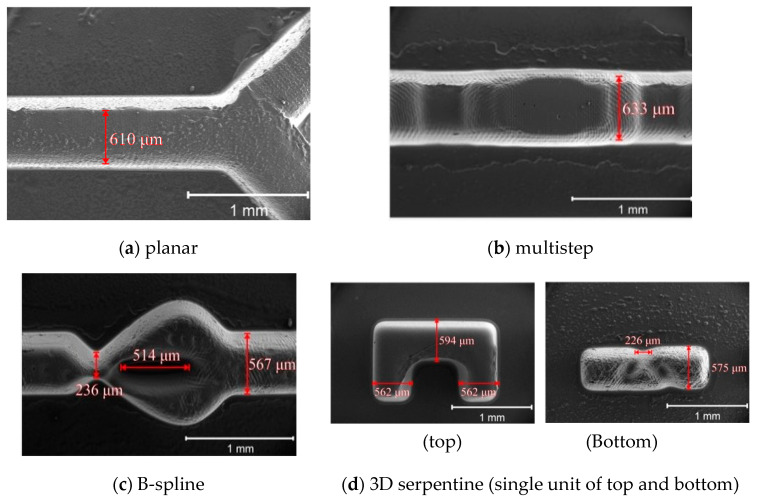
Top view SEM images of micromixers; from top and in clockwise direction: (**a**) planar, (**b**) multistep, (**c**) B-spline and (**d**) 3D serpentine (top and bottom).

**Figure 11 micromachines-11-00548-f011:**
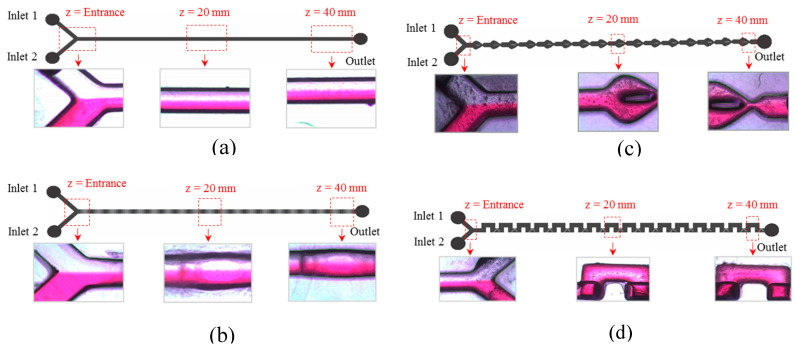
Experimental mixing results at the Y-junction, middle and end of mixing channel in: (**a**) Planar (**b**) Multi-step, (**c**) B-Spline and (**d**) 3D Serpentine micromixers.

**Figure 12 micromachines-11-00548-f012:**
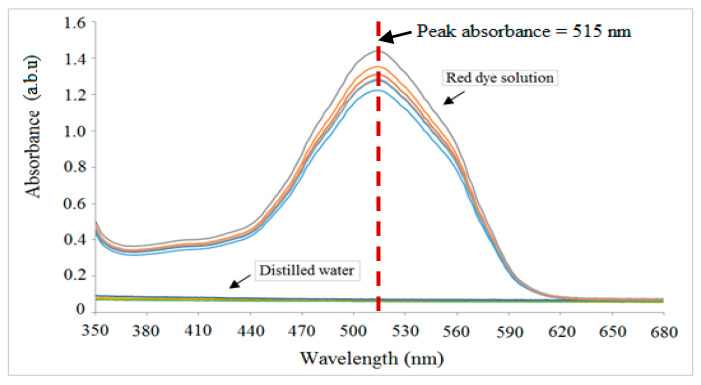
Spectrometry analysis for the highest and lowest concentration solution.

**Table 1 micromachines-11-00548-t001:** Summary of lateral dimension of the fabricated micromixers.

Type	Targeted Width (μm)	Actual Width (μm)	Deviation Error (%)
Planar	600	610	1.7
Multistep	600	633	5.5
B spline	600	567	0.5
3D Serpentine (top) 3D serpentine (bottom)	600 600	562 and 594 575	6.3 and 1.0 4.2

**Table 2 micromachines-11-00548-t002:** Calibration points used to determine the mixing performance.

Concentration of Red Dye	Calculated Percentage of Red Dye (%)	Absorbance Value
Highest concentration (1 mg/mL)	100	1.3 (Spectroscopy measurement)
Complete mixing (0.5 mg/mL)	50	0.65 (Calculated based on Beers law assumption
Lowest concentration (0 mg/L)	0	0.1 (Spectroscopy measurement)

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
