# Peer review of "Implementation of a Single Emulsion Mask for Three-Dimensional (3D) Microstructure Fabrication of Micromixers Using the Grayscale Photolithography Technique"

_micromachines, 2020, doi:10.3390/mi11060548_

Round 1

Reviewer 1 Report

Hamid et al. reported a 3D microfluidic mixer fabricated by gray scale lithography. Gray scale lithography is a useful method for fabricating 3D channels. The present work exploits this method for fabricating passive microfluidic mixer. This work is publishable if the following concerns are properly addressed.

  1. SU8-10 is used for this work. Is there any specific reason for using SU8-10? Many microfluidic applications prefer channels with height much less than 1000 um. It may be worth to look at other types of SU8.
  2. 6b shows some surface roughness. Does the surface roughness affect mixing or the reproducibility of the device?
  3. How a plate reader is combined with an inverted microscope? How the absorbance is measured? On-chip or off-chip? Proper statistical analysis should be performed to determine which design is better. It is impossible to compare any values without replicate trials and confidence intervals.

Author Response

Reviewer 1 report as attachment

Reviewer 2 Report

This is an interesting paper which presents a relatively simple method of manufacturing 3D microfluidic mixer structures using PDMS soft lithography using a master created in SU-8 photoresist. The SU8 master of varying depths is created on an optically transparent substrate (here glass) using backside UV illumination via an emulsion gray scale lithographic mask. This useful and interesting approach has been reported before, e.g. the early reference of 2004 which the authors should therefore reference:

Three dimensional micromachining of SU-8 and application for PDMS micro capillaries, R. Mori et al, 6th International Conference on Miniaturised Systems for Chemistry and Life Sciences, September 26 – 30 2004, Malmo Sweden, p 333.

The degree of mixing for four different mixer types is investigated for the input of distilled water and water containing a red coloured dye – both qualitatively using visual inspection of coloured images and also by measuring the absorbance at a wavelength corresponding to the peak of the absorption by the red dye.

The paper is clear, well written and easy to follow. The authors should address the following points:

  • Please explain the method for gluing the two PDMS layers together.
  • Please describe in more detail the form for the mixers; in particular the B-spine mixer -specific mathematical form.
  • Figure 6 – please make the scale markers clearer
  • Figure 7 – there are five images and four captions – the captions also appear to be displaced. Also please describe how the complete mixer is composed of a number of these individual mixer units in series.
  • Figure 9 – gives the absorbance as a function of wavelength – please state what the red dye is. The absorbance is related to concentration via the Beer (more commonly referred to as Beer-Lambert) law. The absorbance is also related to the path length. Thus when using absorbance to deduce the concentration via absorbance the same path length must be used. Thus, please describe what the pathlength is for this curve.
  • Further description is required as to how the efficiency of mixing is deduced; particularly as the absorbance also depends on the pathlength. Describe precisely what the absorbance values refer to and whether the depth of the channel at which this is measured is the same for all cases.
  • The data which would be of most interest to readers would be the variation of the profile of the concentration of the dye across the exit channel of the mixer device. Thus (a) would show a large step change at the centre of the channel whereas (c) would not. A better quantification of mixing would then be a measure of the variation of absorbance across the exit channel from the mixer.

Author Response

Reviewer 2 report as attachment

Round 2

Reviewer 1 Report

publish as is